# ANALYZING THE ROLE OF SPINAL JOINT DYNAMICS IN THE MOVEMENT OF A SPRAWLING ROBOT

## ABSTRACT

Sprawling locomotion in vertebrates, particularly salamanders, demonstrates how body undulation and spinal mobility enhance stability, maneuverability, and adaptability across complex terrains. While prior work has separately explored biologically inspired gait design or deep reinforcement learning (DRL), these approaches face inherent limitations: open-loop gait designs often lack adaptability to unforeseen terrain variations, whereas end-to-end DRL methods are data-hungry and prone to unstable behaviors when transferring from simulation to real robots. We propose a hybrid control framework that integrates Hildebrand's biologically grounded gait design with DRL, enabling a salamander-inspired quadruped robot to exploit active spinal joints for robust crawling motion. Our evaluation across multiple robot configurations in target-directed navigation tasks reveals that this hybrid approach systematically improves robustness under environmental uncertainties such as surface irregularities. By bridging structured gait design with learning-based methodology, our work highlights the promise of interdisciplinary control strategies for developing efficient, resilient, and biologically informed spinal actuation in robotic systems.

## 1 INTRODUCTION

Among the vertebrates, salamanders, with their unique ability to transition between walking and swimming gaits, highlight the role of spinal mobility in locomotion Phan et al. (2020). A flexible spine enables undulation of the body, a wavelike motion along the spine that supports diverse movements, aiding navigation over uneven terrains and obstacles Lee et al. (2020). Such biomechanical principles have been the inspiration for the development of robotic systems that are capable of mimicking these natural movements and can overcome the limitations of rigid structures, achieving greater efficiency and adaptability in complex environments Phan et al. (2020).

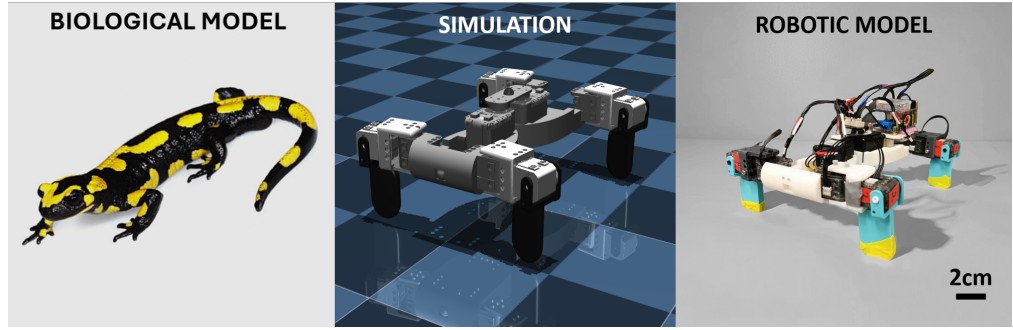

Figure 1: **Biological and robotic model**. (left) The fire salamander (*Salamandra salamandra*) is a common species of salamander found in Europe. (middle) The robot model in the Mujoco simulation world. (right) 3D printed robotic salamander.

In both animals and robots, the coordination between body undulation and limb movement is of high significance for effective locomotion Ijspeert et al. (2007); Chong et al. (2018; 2019); Ozkan Aydin et al. (2017); Suzuki et al. (2021). For salamanders, this coordination is essential for creating

propulsion during swimming and ensuring stability while walking Ijspeert et al. (2007). Replicating this morphology in robotics presents a significant challenge as it requires precise control of multiple degrees of freedom.

Yet environmental uncertainties, such as surface irregularities and variations in friction, can significantly disrupt body-limb coordination and cause discrepancies between predictions from mathematical models and real-world outcomes. Addressing this challenge requires the development of sophisticated control strategies capable of dynamically adapting to uncertain conditions while maintaining efficient locomotion. Building on biomechanical insights, incorporating deep reinforcement learning (DRL) techniques with sensory processing further advances the field of robotic locomotion Kober et al. (2013), enabling more dynamic approaches and greater adaptability to diverse scenarios. DRL offers a promising framework for handling stochastic environments, and adapting to challenging conditions Benbrahim & Franklin (1997); Tedrake et al. (2004); Peng et al. (2015; 2016; 2017); Heess et al. (2017); Fu et al. (2021); Bellegarda et al. (2022); Ji et al. (2022); Aractingi et al. (2023); Han et al. (2024); Hoeller et al. (2024); Bellegarda et al. (2024).

In this study, we comparatively examine learning-based control strategies, a biologically inspired gait design method, and their hybrid framework on a salamander-like robot (Fig. 1). Specifically, we evaluate two distinct robot configurations: one employing a fixed spinal joint and another featuring an active spinal joint. Initially, the salamander robot utilizes a predefined footfall pattern based on the Hildebrand gait method Hildebrand (1985), with a fixed spinal joint facilitating linear locomotion. Subsequently, the robot retains this footfall pattern but incorporates an active spinal joint, which enables dynamic body undulation achieved through DRL-based method. Furthermore, we train and evaluate the robot under various scenarios and compare these results in our experimental section. To enable precise control and facilitate effective learning, we developed a digital twin of the physical robot integrated within the MuJoCo simulation environment Todorov et al. (2012). The main contributions of our work are as follows:

- We present an extensive analysis of crawling locomotion in a salamander-inspired quadruped, which broadens the quadruped robotics literature that predominantly focuses on dog-like morphologies.
- We introduce a hybrid framework that combines the strengths of biologically inspired gait design with deep reinforcement learning, which balances stability and adaptability.
- We conduct a comprehensive experimental evaluation across (1) fixed versus active spinal joints and (2) simulation-to-real transfer to demonstrate the effectiveness of our approach.

## 2 MATERIALS AND METHODS

### 2.1 ROBOT DESIGN & ACTUATION

The robot design is inspired by the biomechanics of the salamander after conducting a comprehensive review of amphibious and sprawling locomotion Ijspeert (2020); Bing et al. (2023). To strike a balance between biological fidelity and mechanical simplicity, we developed a simplified

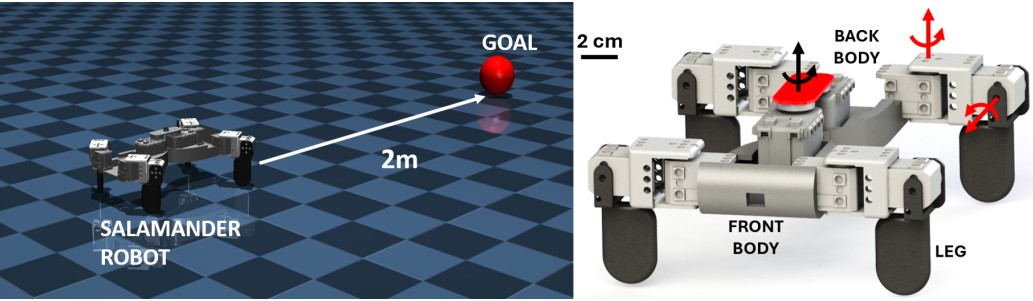

Figure 2: **Simulation Environment and Robot Actuation**. (Left) The MuJoCo simulation environment used for evaluating the robot's locomotion. The red sphere indicates the target goal. (Right) CAD model of the robot detailing its kinematic structure. The red arrows represent the two rotational axes per leg, and the black arrow indicates the single spinal joint that enables lateral bending for sprawling locomotion.

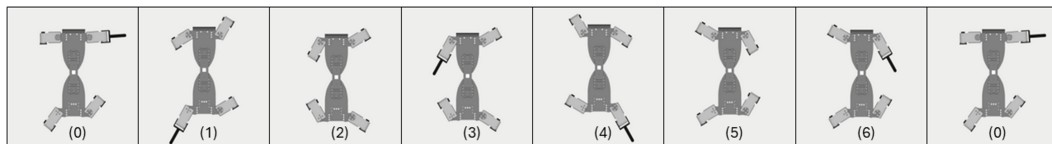

Figure 3: **Example biological gait-cycle.** Joint positions during one walking cycle following the Hildebrand-style gait for both the passive and active spinal joint scenarios. Each leg spends 25% of the cycle in the air and 75% on the ground.

quadrupedal model. Each of the four legs has 2 degrees of freedom (DoF), allowing for controlled motion in the fore-aft (protraction/retraction) and up-down (elevation/depression) planes. To replicate the body undulation characteristic of a sprawling gait, a single 1-DoF spinal joint was integrated into the robot's trunk. The body joint facilitates lateral bending of the front and rear body segments in the horizontal plane. This design effectively captures the core principles of salamander locomotion while maintaining a manageable level of complexity for control and simulation.

The salamander robot is designed using SolidWorks (Fig-2), and mechanical components, including the legs, shoulders, and front and rear body segments, are 3D printed using ABS filament with a Stratasys F170 3D printer. The leg movements are powered by Dynamixel XL-320 servo motors, while a Dynamixel AX-12 servo actuates the spinal joint. These servos provide real-time feedback on both torque and position. A U2D2 Power Hub Board handles power distribution, using TTL and RS-485 communication to connect the servo motors to the controller. This design, visually represented in Fig-2, effectively captures the essential biomechanics of salamander movement.

## 2.2 Open-Loop Hildebrand Gait Design Method

In this study, we utilize the Hildebrand gait analysis method, a well-established classification system for characterizing quadrupedal locomotion based on the timing and sequencing of footfalls Hildebrand (1977; 1965). This method has been widely adopted in both biomechanical studies of animals and the control of legged robots Chong et al. (2019; 2021). Hildebrand's framework defines symmetrical gaits using two key kinematic parameters: the duty factor and the leg phase shift. The duty factor ($\beta$) is the percentage of a full gait cycle during which a single foot is in contact with the ground. Gaits with a duty factor greater than 0.5 are classified as walks, as at least two feet are always on the ground, ensuring static stability. The leg phase shift ($\phi$) is the temporal offset, expressed as a percentage of the gait cycle, between the footfalls of ipsilateral (same-side) limbs. For example, a phase shift of 0.5 (or 50%) represents a trot-like gait where diagonal pairs of legs move in unison.

In this work, we employed a specific walking gait characterized by a duty factor of 0.75, meaning each leg spends 75% of the cycle in the stance phase and 25% in the swing phase. This high duty factor ensures continuous ground contact and inherent stability. The leg phase shift was set to 0.25 (25%), corresponding to a lateral-sequence gait, where the footfall sequence on one side of the body is hind-fore, then the same sequence on the other side. This gait pattern is particularly common in sprawling animals like salamanders and offers a balance between stability and maneuverability. Figure 3 illustrates the commanded joint angles for each leg throughout a single gait cycle. This cyclic pattern is continuously iterated as the robot navigates toward its target.

## 2.3 Deep Reinforcement Learning

Reinforcement learning (RL) provides a general framework for solving sequential decision-making problems, which underpins much of robot learning Sutton et al. (1998); Schulman et al. (2015a); Gu et al. (2017); Haarnoja et al. (2018); Fujimoto et al. (2018); Li et al. (2025). In line with prior work on quadruped robots, we train our salamander-inspired robot using both PPO and SAC. Since SAC consistently outperforms PPO in our experiments, we report results using SAC. Soft Actor-Critic is an off-policy RL algorithm that extends actor-critic methods by incorporating entropy regularization

to incentivize exploration:

$$\pi^* = \arg\max_{\pi} \mathbb{E}_{\tau \sim \pi}\left[\sum_{t=0}^{\infty} \gamma^t \left(R(s_t, a_t, s_{t+1}) + \alpha H\left(\pi(\cdot|s_t)\right)\right)\right] \tag{1}$$

where

$$H(X) = -\sum_{x \in \mathcal{X}} P(x) \log P(x) \tag{2}$$

represents the entropy term. Here, $\alpha > 0$ is a trade-off parameter that balances the reward function and the entropy term, and hence exploration and exploitation Mnih et al. (2016); Lillicrap et al. (2015); Schulman et al. (2015b); Haarnoja et al. (2018). The SAC algorithm is capable of handling high dimensional observation and continuous action spaces while maintaining stability during learning.

| STATE VARIABLES | 8-JOINTS | 9-JOINTS |
|---|---|---|
| X/Y/Z COORDINATES OF TORSO | 3 | 3+3 |
| X/Y/Z/W ORIENTATIONS OF TORSO | 4 | 4+4 |
| LIMB JOINT ANGLES | 2x4 | 2x4 |
| X/Y/Z VELOCITIES OF TORSO | 3 | 3+3 |
| X/Y/Z ANGULAR VELOCITIES OF TORSO | 3 | 3+3 |
| ANGULAR VELOCITIES OF LEG JOINTS | 2x4 | 2x4 |
| SPINAL JOINT ANGLE | 0 | 1 |
| SPINAL ANGULAR VELOCITY | 0 | 1 |
| DISTANCE TO GOAL | 3 | 3 |
| ANGLE TO GOAL | 1 | 1 |
| FRICTIONS (X/Y/Z) ON LEGS | 3x4 | 3x4 |
| TOTAL | 45 | 60 |

Figure 4: **State variables and Results.** (Left) Observation space for the 8-joint and 9-joint robot configurations. (Right) RL Learning Curves.

Fig-4 (left) presents the observation space for both the 8-joint (no spinal joint) and 9-joint robot configurations, while the action space consists of joint angles (in radians) in our setup. The reward function is inspired by MuJoCo's ant environment and previous work done on quadruped robots Boney et al. (2022); Tan et al. (2018) to encourage efficient moves towards the goal while maintaining stability. It consists of multiple components:

$$R(s, a) = w_1 \cdot \Delta x + w_2 \cdot \Delta d + w_3 \cdot \Delta y + w_4 \cdot C + H \tag{3}$$

where $\Delta x$ represents the change in the x-direction to incentive forward movement, $\Delta d$ indicates the change in distance to the goal, $\Delta y$ is the change in the y-direction to incentivize the stability and ensure smooth movement, $C$ is control cost term to punish the robot for taking excessively large actions, and $H$ represents the constant healthy reward (to prevent the robot from jumping or flipping), and $w_1 = 5, w_2 = 150, w_3 = 5, w_4 = 0.1$ are the corresponding weights which are tuned for the best learning.

The hyperparameters used for training the reinforcement learning model are listed in Table 1. These include the batch size, learning rate, number of steps collected before learning begins, discount factor (which balances short-term vs. long-term rewards), entropy coefficient (which controls exploration-exploitation trade-off), and soft update coefficient (which governs the update rate of the target network). All parameters are tuned to optimize learning stability and performance.

## 3 EXPERIMENTAL SETUP & RESULTS

The simulation environment is built using MuJoCo Todorov et al. (2012). During simulation, the robot employs Hildebrand gaits or DRL actions to locomote toward a goal, a red ball positioned 2 meters away from the initial position (Fig.2).

The gaits exhibit cyclic behavior, i.e. repeats at a fixed period. However, this periodicity does not disrupt forward motion,and advance the robot smoothly toward the target. At each discrete time point, joint angles (in radians) are prescribed to the robot's actuators, and forward kinematics calculations update its position. The simulation concludes once the robot reaches the goal after a predetermined number of iterations.

In this section, we assess the efficiency of our hybrid RL–Hildebrand framework and try to answer two core questions:

1. How does end-to-end RL compare to classical Hildebrand gaits in terms of goal accuracy and stability?

2. What is the impact of an active spinal joint versus a fixed spine on overall crawling locomotion performance?

To this end, we evaluate all methods on a standardized target-directed navigation task and conducted a series of comparative studies, testing four distinct locomotion policies: (1) a traditional Hildebrand-only gait, (2) an unconstrained reinforcement learning (RL) policy, (3) an RL policy with torque constraint, and (4) our proposed hybrid RL–Hildebrand framework with an actively controlled spinal joint. The objective was to systematically assess how spinal flexibility and the integration of biologically inspired priors influence locomotion robustness and performance.

Experiments were conducted over 100 simulated episodes, and the average results are reported. The comparison of the models is performed in the following way by using three key metrics: (1) **MDB:** the minimum distance to the goal over the full trajectory. (2) **DY (Lateral Deviation):** the maximum displacement along the y-axis over the full trajectory. (3) **ATB:** the average timesteps to reach the goal.

Table 1: Hyperparameters used to train SAC.

| Hyperparameters | 8-joints | 9-joints |
|---|---|---|
| Batch size | 128 | 128 |
| Learning starts | 5000 | 5000 |
| Discount factor ($\gamma$) | 0.958 | 0.972 |
| Learning rate | 0.0007 | 0.0053 |
| Entropy coefficient ($\alpha$) | 0.0815 | 0.04 |
| Soft update parameter ($\tau$) | 0.006 | 0.016 |

An episode is marked as successful if

$$\text{MDB} \leq 0.15 \quad \text{and} \quad \text{DY} \leq 0.05. \tag{4}$$

For successful episodes, ATB quantifies traversal efficiency. While a lower ATB indicates faster navigation, we emphasize that excessively rapid trajectories may be infeasible on the physical salamander-inspired robot, and hence is not necessarily better in practice.

### 3.1 SAC vs Hildebrand Gaits for 8-joint Robot Configuration

Our experimental evaluation first aimed to address (1) How does an end-to-end reinforcement learning (RL) policy compare to classical Hildebrand gaits in terms of goal accuracy, speed, and stability? To investigate this, we conducted a comparative study of the 8-joint robot, which lacks an active spinal joint, using three distinct control strategies: a traditional Hildebrand gait, an unconstrained RL policy, and a torque-limited RL policy (denoted as RL*, during inference, the actions/angles obtained from the RL model are clipped by applying torque limits to the motors).

Table 2 illustrates the performance of the robot under these three conditions. The Hildebrand gait, while producing robust and easily executed motions, yielded relatively slow, walking-like locomotion. In contrast, an unconstrained RL agent learns high-speed, cheetah-like gaits that exceed the salamander robot's

Table 2: Performance comparison of different approaches for the 8-joint configuration. Values are presented as mean $\pm$ standard deviation over 5 seeds. Metrics: MDB (minimum distance to the ball), DY (deviation from the y-axis), and ATB (average timesteps to goal).

| 8-joints Version | MDB | DY | ATB |
|---|---|---|---|
| Hildebrand | 0.10 | 0.03 | 6103 |
| RL | 0.04 ± 0.002 | 0.02 ± 0.001 | 104 ± 1 |
| RL* | 0.13 ± 0.039 | 0.02 ± 0.001 | 334 ± 17 |

Table 3: Performance comparison of different approaches for the 9-joint configuration. Values are presented as mean $\pm$ standard deviation over 5 seeds. Metrics: MDB (minimum distance to the ball), DY (deviation from the y-axis), and ATB (average timesteps to goal).

| Method | MDB | DY | ATB |
|---|---|---|---|
| Hildebrand + RL | $0.05 \pm 0.001$ | $0.03 \pm 0.0008$ | $1518 \pm 6$ |
| RL | $0.07 \pm 0.008$ | $0.04 \pm 0.001$ | $196 \pm 2$ |
| RL* | $0.10 \pm 0.004$ | $0.05 \pm 0.002$ | $1931 \pm 1$ |

physical capabilities. While this behavior was highly effective in the idealized simulation, it is not physically realizable on our hardware. To bridge this gap between simulation and reality, we introduced actuator force limits ($[-0.39, 0.39]$) to the RL policy, creating the $RL^*$ strategy. This simple constraint successfully curbed excessive velocities, resulting in a policy that was both significantly faster than the Hildebrand baseline and more stable than the pure RL policy.

### 3.2 FIXED VS ACTIVE SPINAL JOINT ROBOT CONFIGURATIONS

We address the second core question: what is the impact of an active spinal joint versus a fixed spine on overall locomotion performance? As shown in Figure 5, an active spinal joint significantly expands the robot's kinematic workspace. For identical leg joint angles, the 9-joint configuration with an active spine achieves a greater forward reach than the 8-joint configuration with a fixed spine. This demonstrates that adding an active spinal joint fundamentally enhances the robot's propulsive capabilities and maneuverability.

Table 3 compares the performance of the 9-joint robot across three distinct control strategies: a hybrid model (Hildebrand

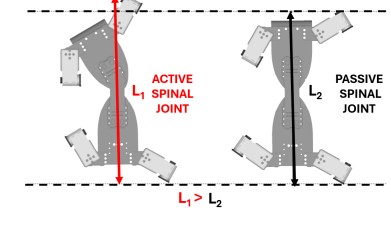

Figure 5: **Reachability with and without an Active Spinal Joint.** The figure compares the forward reachability of the robot with a flexible spine (9-joints) and a rigid spine (8-joints). As shown, the active spinal joint enables the robot to extend its reach farther forward ($L_1 > L_2$) for the same leg joint angles.

gaits for legs + RL for the spinal joint), a pure end-to-end RL policy for all joints, and a torque-constrained RL policy ($RL^*$) for all joints.

Unlike the 8-joint configuration, where imposing torque limits on a pure RL policy led to improved locomotion, directly constraining the end-to-end RL policy on all 9 joints did not yield a stable or efficient gait. Instead, we found that the most effective strategy was our hybrid approach, which uses Hildebrand gaits to control the rhythmic leg movements and an RL policy to adaptively control the spinal joint. This produced a more efficient and physically feasible locomotion pattern for our salamander-inspired robot. These results highlight the strong potential of combining biologically inspired gait design methods with learning-based controllers to achieve robust and adaptable robotic locomotion.

### 3.3 LOCOMOTION ON ROUGH TERRAIN

To generate the rough terrain used in our simulations, we procedurally synthesized a heightmap by combining multiple octaves of Gaussian-filtered random noise. Each octave was normalized and scaled according to persistence and lacunarity parameters, allowing us to capture both large-scale elevation changes and fine-grained irregularities.

We evaluated the performance of several strategies on two distinct terrain levels, categorized as easy and hard (Table 4, Fig. 6). Similarly, we compared three control strategies: an 8-joint RL model, a 9-joint RL model (with an added joint for spinal control), and a combined Hildebrand + RL approach. For both the 8-joint RL and 9-joint RL models, we also tested a torque-limited

version denoted with an asterisk (*). Performance was quantified using two metrics, MDB and ATB, described in Section 3. The experimental results on rough terrain support the effectiveness of our hybrid approach. Among all methods, only the 8-joint RL baseline and our hybrid approach consistently reach closer to the target ball.

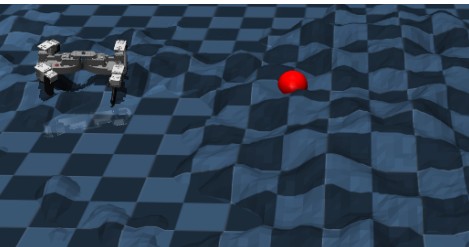

Figure 6: Salamander robot on rough terrain

Table 4: Performance on two levels of terrain.

|  | Easy Level Terrain | | Hard Level Terrain | |
| --- | --- | --- | --- | --- |
|  | MDB | ATB | MDB | ATB |
| 8-Joint RL | 0.40 | 540 | 0.57 | 350 |
| 8-Joint RL* | 0.77 | 449 | 1.35 | 217 |
| 9-Joint RL | 0.31 | 307 | 0.66 | 946 |
| 9-Joint RL* | 1.41 | 1002 | 2.31 | 1002 |
| Hildebrand + RL | 0.49 | 1674 | 0.29 | 3358 |

## 3.4 SIM-TO-REAL GAP: OBSERVATIONS

While simulation experiments provided valuable insights regarding locomotion strategies, physical implementation on the salamander-inspired robot revealed limitations that highlight the sim-to-real gap. For both 9-joint (active spine) and 8-joint (rigid spine) configurations, reinforcement learning policies trained in simulation, whether unconstrained (RL) or torque-limited (RL*), proved infeasible on hardware. These policies generated high-frequency, cheetah-like gaits that exceeded the torque and velocity capabilities of the physical robot's actuators. Even torque-limited policies produced oscillatory, unstable motions, confirming that behaviors optimized in idealized environments did not generalize to the physical platform.

Thus, only two locomotion plans proved appropriate for real-world test: Hildebrand gait in the 8-joint configuration and Hildebrand+RL hybrid framework with Hildebrand gait controlling the legs and RL policy controlling the spinal joint in the 9-joint configuration. This limitation defines the importance of incorporating biologically inspired priors in control policies to make them physically realizable.

Real-world trials showed that the hybrid Hildebrand+RL configuration achieved an average of 38% faster traversal speed relative to the Hildebrand-only baseline, while maintaining stable, goal-directed locomotion. This improvement was primarily reflected in the increased forward displacement over the same time interval, highlighting the contribution of active spinal control to overall locomotor speed.

As shown in Figure 7(a), the hybrid configuration achieved a forward displacement of $150.4\pm2.1$ cm in the same 24-second interval where the Hildebrand-only gait reached $108.6\pm5.4$ cm. Figure 7(b) further quantifies this difference, showing consistently greater forward progress over time. These plots correspond to a single representative trial, included for illustrative purposes, while the averaged results across repeated trials are reported in Table 5. Lateral deviations observed in both cases stem from the sim-to-real gap, including differences in weight distribution between the robot and its digital twin, unmodeled surface interactions, and reliance on open-loop actuation.

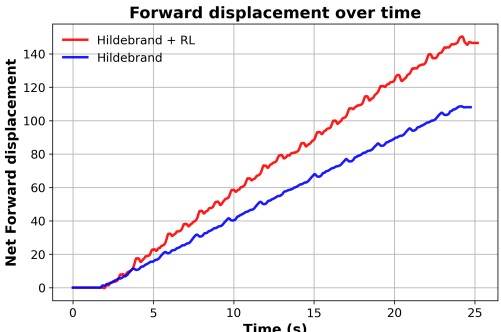

(a) Trajectories from motion capture. Hybrid Hildebrand+RL (red): 150.4 cm; Hildebrand-only (blue): 108.6 cm.

(b) Forward displacement over time for the same trial, showing a 38% speed increase.

Figure 7: Comparison of real-world performance: (a) trajectories and (b) forward displacement over 24 seconds.

Table 5: Comparison of real-world locomotion performance. Values show mean forward displacement and average speed after 24 seconds (averaged over repeated trials).

| Method | Forward displacement (cm) | Speed (cm/s) |
|---|---|---|
| Hildebrand (8-joint) | $108.6 \pm 5.4$ | 4.54 |
| Hybrid Hildebrand+RL (9-joint) | $150.4 \pm 2.1$ | 6.27 |

In addition to flat-ground trials, we conducted preliminary qualitative tests on uneven terrain, including rocks (2-4 cm) and dry sand. In both cases, the hybrid Hildebrand+RL strategy enabled successful traversal, as illustrated in Figure 8. These results indicate that active spinal control contributes to robustness by redistributing its weight to maintain stability and traction, beyond controlled laboratory conditions. A more systematic quantitative evaluation of locomotion across varied terrains will be pursued as part of future work.

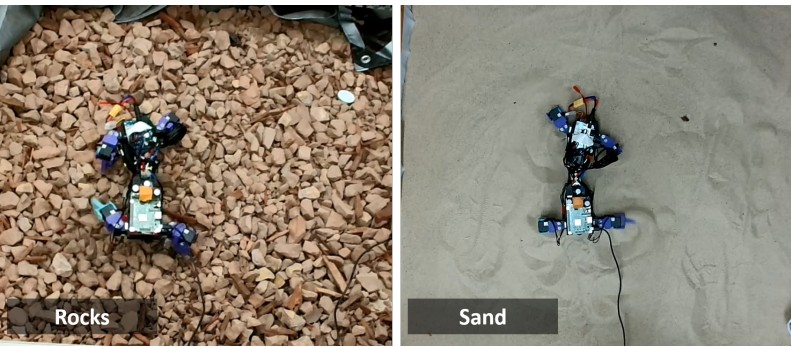

Figure 8: Qualitative evaluation of the robot on uneven terrain. The hybrid Hildebrand+RL strategy enabled successful traversal on (left) rocks (sizes between 2-4 cm) and (right) dry sand.

These findings confirm that RL policies trained in a purely simulated environment are not directly transferable to the physical salamander robot. This discrepancy may arise because standard RL simulations, while accounting for some physical constraints, frequently converge on gaits optimized for idealized point-contact dynamics and upright-legged platforms like Unitree robots. Such policies are fundamentally incompatible with the sprawled-legged anatomy of the salamander robot, where the limbs are in continuous ground contact and exhibit pronounced sweeping motions during the stance phase.

In contrast, our results demonstrate that a carefully structured hybrid approach can successfully mitigate a significant portion of the sim-to-real gap by explicitly accounting for the unique kinematic

and dynamic properties of the sprawled architecture. At the same time, they highlight unresolved challenges that motivate our future research directions, including more systematic terrain evaluation and biologically inspired control priors.

### 3.5 FUTURE WORK AND CONCLUSION

A primary avenue for future research lies in refining the reward function, a critical component of reinforcement learning that significantly influences the learned behavior. The hand-crafted reward schemes commonly used in quadrupedal robotics, which often penalize energy consumption while rewarding forward velocity, can inadvertently lead to undesirable or unsafe behaviors. To address this, we propose leveraging formal specifications through Signal Temporal Logic (STL) to systematically and more safely shape the reward function Balakrishnan & Deshmukh (2019); Kulgod et al. (2020). This STL-based approach provides a principled and robust alternative to heuristic reward design, offering greater sample efficiency and a more formal guarantee of desired robot behaviors.

While our current RL policy proved effective in simulation, a significant challenge remains in bridging the sim-to-real gap, where unmodeled dynamics and hardware constraints can render a simulated policy unusable on a physical robot. The aggressive and unstable motions generated by our current policy highlight this issue. To overcome this limitation, our future work will focus on integrating Central Pattern Generators (CPGs) as a biologically inspired control prior. CPGs are neural networks found in animals that can generate rhythmic motor commands without continuous sensory feedback Ijspeert (2008), making them ideal for producing stable and coordinated gaits in robots. Our proposed solution would not replace the RL agent but would instead use it to modulate high-level CPG parameters, such as frequency, phase offset, and amplitude. This approach preserves the dimensionality of the control space while embedding structured, rhythmic patterns that are inherently more respectful of the robot's physical limitations. By delegating low-level timing and coordination to the CPG, the RL agent can operate in a more abstract state space, which is expected to enhance training efficiency and stability, particularly in noisy or partially observable environments. We aim to train this hybrid CPG–RL system in simulation and subsequently validate its performance on our physical salamander-inspired robot across diverse terrains.

In conclusion, our study demonstrates that the ability to integrate biologically inspired models, such as the Hildebrand gait, into learning-based methods represents a promising research direction. This interdisciplinary approach not only offers new insights into creating more natural and efficient locomotion for robotic systems but also highlights the potential of combining principled, model-based priors with the adaptability of data-driven learning. Ultimately, this work serves as a foundation for developing more robust and versatile control systems capable of navigating complex, real-world environments.

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
