# OpenReview forum: "Analyzing the Role of Spinal Joint Dynamics in the Movement of a Sprawling Robot"
_ICLR.cc/2026/Conference — ICLR 2026 Conference Withdrawn Submission_

### Official Review · Reviewer_az6P · 2025-10-28

**Soundness:** 2
**Presentation:** 3
**Contribution:** 2
**Rating:** 2
**Confidence:** 3

**Summary:**

The paper investigates how active spinal joint dynamics influence the locomotion performance of sprawling quadruped robots inspired by salamanders. To this end, the authors design a salamander-like quadruped robot with four legs (each with two degrees of freedom) and an optional one-degree-of-freedom spinal joint for lateral bending. The robot’s legs are controlled using a biologically inspired Hildebrand gait, while the spinal joint is adaptively modulated by a deep reinforcement learning (DRL) policy to improve locomotion efficiency and robustness. The experiments compare four locomotion strategies—Hildebrand-only, RL-only, torque-limited RL (RL*), and the hybrid Hildebrand + RL approach—in both simulation (MuJoCo) and real-world hardware. The results demonstrate that the hybrid Hildebrand + RL framework achieves the best balance between stability and adaptability, outperforming both purely model-based and purely learning-based baselines in reaching the target efficiently.

**Strengths:**

- Biologically grounded motivation with strong interdisciplinary reasoning.
- The writing is clear and well-organized, making the technical content easy to follow for readers from both robotics and machine learning backgrounds.
- The experiments are comprehensive, demonstrating results in both simulation and real-world hardware, which strengthens the paper’s practical relevance and credibility.

**Weaknesses:**

Novelty: The paper’s use of a hybrid controller combining a biologically inspired gait (CPG/Hildebrand) with deep reinforcement learning is not clearly novel. For example, CPG‑RL: Learning Central Pattern Generators for Quadruped Locomotion (Bellegarda & Ijspeert, 2022) already integrates central pattern generators with RL for quadruped locomotion.The paper under review should better articulate how its contribution exceeds or differs from prior CPG + RL frameworks.

Results are not fully convincing: In the current era, complex quadruped systems can be trained in simulation and transferred to real hardware with relative success.Thus it is hard to be convinced that the relatively simpler robot system in this paper faces major sim-to-real difficulty, or that the added spinal joint truly drives the transfer gap.

Ambiguity in focus: The paper’s title and introduction suggest an analysis of spinal joint functionality (inspired by animal biology) via combining biomechanical insight and learning. Yet in the current version the emphasis appears to be mostly on “how using RL to control an extra DoF (spinal joint) affects policy training/performance.” This mismatch creates unclear expectations: is the goal biological insight or robotic control improvement? The clarity of the paper’s scope and claims would benefit from refining.

**Questions:**

N/A

---

> ### Author Response · Authors · 2025-12-01
> **Thanks to the Reviewer**
>
> We thank the reviewer for their constructive feedback and appreciate the opportunity to clarify how our work differs from existing CPG–RL frameworks and to highlight the specific challenges of sprawling locomotion.
> * Prior CPG-based controllers are often formulated via symmetric MDPs with an action space over phase variables, which fixes the gait structure and can limit adaptability. In our setting, the policy operates directly in joint space: we use RL to control the lateral undulation of the spine while the limbs follow a Hildebrand pattern. This is not simply applying RL to tune a CPG; it uses RL to learn the nonlinear coupling between a flexible spine and rigid limbs in a sprawled configuration. To the best of our knowledge, this particular decomposition (fixed limb gait + RL spine modulation) has not been systematically studied in existing work.
>
> * We argue that mechanical simplicity does not imply dynamical simplicity. The sprawled posture and lateral spine create contact interactions that are harder to model accurately than, for example, standard trotting gaits on stiff, upright quadrupeds without spine joint. Our robot uses a salamander-like, sprawled morphology with a lateral spine rather than a dog-like quadruped, leading to qualitatively different contact patterns and failure modes, which we explicitly study in the context of sim-to-real transfer.
>
> * We apologize for any ambiguity in the current presentation. Our primary goal is robotic control improvement informed by biological insight. We will revise the title, abstract, and introduction to make this focus explicit and to clearly state that the central contributions are (i) a hybrid control architecture leveraging biological priors and (ii) an analysis of how an actuated spine and controller structure affect learned locomotion and sim-to-real performance.

---

### Official Review · Reviewer_acjD · 2025-10-31

**Soundness:** 2
**Presentation:** 3
**Contribution:** 2
**Rating:** 4
**Confidence:** 4

**Summary:**

This paper studies locomotion control in a salamander-inspired quadruped robot with an articulated spine joint in addition to 8 leg joints (2 per leg).  Several methods are compared: A classical Hildebrand-style open-loop trajectory, a pure RL controller, and a hybrid that uses a Hildebrand trajectory for the legs but RL for the spine.  Experiments include both simulation and hardware execution, and both smooth and rough terrain in both cases.  The results suggest that the hybrid approach is particularly useful in the presence of rough terrain.

**Strengths:**

- To my knowledge, an articulated spine is an interesting and under-explored form factor for quadruped robots.
- Overall the paper is clearly written and easy to follow.
- The paper considers a fairly wide range of different experimental conditions, and averages performance over multiple runs to bolster reproducibility.

**Weaknesses:**

- I am not sure how relevant this paper is to ICLR, which is focused on deep learning architectures.  This paper is primarily focused on robotics, not deep learning.  It does use reinforcement learning, but does not define the network architecture of the agent and value function, nor does it analyze their learned representations.  The robot form factor and hybrid controller may be novel, but as far as I can tell, there is no novelty in terms of the deep learning architectures used (assuming they used standard MLPs common in PPO and SAC).
- The paper reports difficulties with sim-to-real transfer, but does not mention using any domain randomization during training, which is standard practice to improve robustness.  Also, if I understood correctly, torque limits are applied at inference time but not during training, which introduces an unnecessary distribution shift that might negatively impact performance.  This raises a concern that the results may not reflect best practices in RL and might be biased by sub-optimal implementation.
- The paper reports overall task performance using the actuated spine, but would benefit from a deeper analysis of the spine behavior, since that is the primary novelty of the work.  In particular, it would be interesting to plot the trajectory of the spine over time, quantify its amplitude and periodicity, etc.
- There are some minor presentation issues that could be improved:
    - The authors should use \citep for citations where the author name is not also a noun in the sentence, so that the parentheses enclose the author name
    - Lines 160-161: The full names of SAC and PPO should be introduced immediately before their first occurrences, with citations for each
    - For better image quality, Figure 4 should use a latex table and vector graphics image instead of two raster images
    - Line 217 is missing a space after a comma

**Questions:**

- The paper mentions that various hyper parameters were "tuned to optimize performance," but not how the tuning was done.  Was it manual? A grid search, and if so, over what ranges? Using a software library like optuna?  The answers should also be included in the main text.
- I am surprised the RL performance degraded so significantly when introducing only one additional action dimension for the spinal joint.  Can the authors provide any evidence-based insight into why this effect was observed?  Is it possible that with more tuning or a different architecture the pure RL would work better?

---

> ### Author Response · Authors · 2025-12-01
> **Thanks to the Reviewer**
>
> **For Weaknesses:**
>
>
> We thank the reviewer for the time spent evaluating our manuscript and for their constructive comments. We certainly agree that our contribution does not lie in the novelty of the learning algorithm itself. Our primary contribution is focused on the learning-based control architecture and the detailed analysis of how spinal joint dynamics interact with Reinforcement Learning (RL) policies. This approach is directly in line with ICLR’s interest in RL for embodied agents.
>
>
> * As noted in the paper, standard "black box" RL (using SAC) fails when applied directly, converging to high-frequency, "cheetah-like" gaits in simulation that are physically infeasible on real hardware. This result highlights a critical limitation in applying current end-to-end learning-based control to non-standard morphologies. Our novelty lies precisely in the Hybrid Control Framework. By integrating a biological prior (Hildebrand gait) to robustly handle the rhythmic limb coordination, and delegating the RL agent to control only the active spinal dynamics, we achieve two things:
> 1. We solve the sim-to-real gap by structuring the control and policy space in a principled way.
> 2. We constrain the RL problem, making the learned solution physically feasible and robust.
>
>     This work directly addresses a core, ongoing question in the ICLR community: How do we effectively inject domain knowledge (priors) into learning algorithms to achieve robust real-world performance? We empirically demonstrate that structured learning (our Hybrid method) succeeds where end-to-end learning fails, offering a generalizable approach for developing controllers for complex physical robots.
>
> * Our robot uses a salamander-like, sprawled morphology with a lateral spine, rather than a standard dog-like quadruped. This leads to qualitatively different contact patterns and failure modes, which we explicitly study in the context of sim-to-real transfer.
> * We thank the reviewer for the suggestion regarding deeper analysis of the spine behavior. In the revised manuscript, we will include quantitative trajectories of the spinal joint to make the learned spinal motion more explicit.
>
>
> **For Questions:**
>
>
> 1. We used Optuna for hyperparameter search.
>
> For **SAC**, we searched over: batch size ∈ {64, 128, 256}, learning_starts ∈ {500, 1000, 5000}, gamma ∈ [0.90, 0.9999] (log-uniform), learning rate ∈ [1e−6, 1e−2] (log-uniform), entropy coefficient ∈ [1e−3, 1] (log-uniform), tau ∈ [0.001, 0.02] (step 0.005)
>
>
> For **PPO**, we searched over: batch size ∈ {64, 128, 256}, n_steps ∈ {512, 1024, 2048}, clip_range ∈ {0.1, 0.15, 0.2}, n_epochs ∈ {3, 5, 7}, learning rate ∈ [1e−6, 1e−2] (log-uniform), entropy coefficient ∈ [1e−3, 1] (log-uniform). We will move these details to the Appendix and refer to them from the main text.
>
> 2. We thank the reviewer for this insightful observation and for engaging critically with our results. We agree in principle that introducing only one additional action dimension, specifically for the spine, should not inherently degrade the ultimate performance of an RL agent, provided the policy architecture and training regimen are optimally configured and sufficiently tuned.
> Our empirical analysis was not intended as a definitive claim that no RL configuration could successfully solve the complex 9-joint locomotion problem under the defined constraints. Rather, the purpose of presenting the Standard SAC results was to serve as a principled baseline for the following two points:
> * To document what a standard, widely-adopted RL pipeline (SAC) learns when applied directly to our novel, non-standard morphology and reward structure.
> * To empirically demonstrate, through direct comparison, the substantial benefit of the proposed hybrid control structure, which achieved robust real-world performance by structuring the policy search space through the inclusion of a biological prior.
> Therefore, the degradation in performance observed with the pure SAC baseline merely highlights the sensitivity of standard RL to morphological novelty and high-dimensional action spaces in the absence of targeted policy constraints, thereby validating the utility of our hybrid approach.
>
> For the degradation, please consider two interacting factors:
> * In a sprawling quadruped, the spinal joint physically couples the front and rear girdles. It is not just an "extra"  degree of freedom; its movement fundamentally changes the required phase relationship between the front and rear legs to maintain balance.
> * Reward and morphology encourage aggressive use of the spine. The reward strongly favors fast forward progress and proximity to the goal, with only modest penalties for large actions. When the spinal joint is added, SAC quickly discovers gaits that exploit high-amplitude spinal oscillations to generate motions that are highly effective in simulation but unstable and non-transferable to the hardware, as we also observe qualitatively in Sec. 3.4.

---

### Official Review · Reviewer_qepU · 2025-11-01

**Soundness:** 2
**Presentation:** 1
**Contribution:** 1
**Rating:** 2
**Confidence:** 4

**Summary:**

This paper investigates the functional role of active spinal joints in sprawled quadruped locomotion, inspired by salamanders, through a hybrid control framework that combines biologically grounded gait design (Hildebrand method) with deep reinforcement learning (DRL). The authors develop a 9-DoF salamander-like robot, modeled and 3D-printed based on amphibian biomechanics, and compare locomotion across several control strategies—pure Hildebrand gaits, unconstrained DRL, torque-limited DRL, and a hybrid Hildebrand+RL approach.

The study evaluates locomotion performance on flat and rough terrains within MuJoCo simulation and partially on the physical robot, focusing on stability, goal accuracy, and traversal speed. Results demonstrate that pure RL policies produce unrealistic “cheetah-like” behaviors unsuited to the physical platform, while the hybrid model achieves robust and efficient crawling with ~38% faster traversal speed in real-world trials.

**Strengths:**

* The salamander robot platform is novel

**Weaknesses:**

* Scope of contribution: The paper’s primary contribution lies more in the robotics and bio-inspired locomotion domain than in advancing the machine learning methodology itself. The deep reinforcement learning component relies largely on off-the-shelf algorithms (e.g., SAC) without introducing novel learning techniques or insights that would meaningfully impact the learning community.
* Similar salamander-inspired or sprawling robot designs have been explored in prior literature, and RL-based locomotion frameworks for such morphologies are already well established. To further strengthen the work, the authors could consider integrating GPU-accelerated simulation or parallel training pipelines to improve policy robustness and sample efficiency, following approaches such as [1].

[1] Qu, Tomson, et al. "Versatile Locomotion Skills for Hexapod Robots." 2024 IEEE/RSJ International Conference on Intelligent Robots and Systems (IROS). IEEE, 2024.

**Questions:**

See weakness

---

> ### Author Response · Authors · 2025-12-01
> **Thanks to the Reviewer**
>
> We thank the reviewer for this insightful comment. We agree that salamander-inspired and sprawling morphologies, as well as RL-based locomotion frameworks, have been explored in prior literature, and we will clarify our positioning accordingly. As noted in the paper, much of the existing work either focuses on CPG-based controllers (often formulated via symmetric MDPs with learned phase variables) that fix the gait structure and limit adaptability, or on end-to-end RL, which is typically demonstrated on more standard dog-like quadrupeds. To the best of our knowledge, there is comparatively little work that (i) studies a salamander-like sprawling quadruped with an actuated spine and (ii) performs a systematic comparison of different control strategies (open-loop gait, unconstrained RL, torque-limited RL, and a hybrid leg–spine factorization) on both flat and rough terrain, as we do.
>
>
> Regarding the suggestion on GPU-accelerated simulation and parallel training: our current implementation already leverages MuJoCo XLA (MJX), a JAX-based implementation of MuJoCo, which enables highly parallelized simulation entirely on the GPU. We will make this significantly clearer in the revised version. The paper cited in [1] is highly complementary to our work: it targets spider-like morphology, while our focus is on ablation over control structure and spinal actuation for a salamander-like sprawling quadruped. We will explicitly cite [1] in the related work section.

---

### Official Review · Reviewer_krbK · 2025-11-01

**Soundness:** 3
**Presentation:** 3
**Contribution:** 3
**Rating:** 8
**Confidence:** 2

**Summary:**

This work introduces a hybrid control framework that combines Hildebrand’s biologically inspired gait design with deep reinforcement learning to enable robust, salamander-like crawling in quadruped robots. Experiments across various robot configurations show that this approach enhances stability and adaptability under environmental uncertainties, demonstrating the benefits of integrating structured gait design with RL.

**Strengths:**

The paper proposes a novel hardware platform and demonstrates the benefits of its hybrid approach by showing positive transfer from sim-to-real while also outperforming classical techniques. The paper is well written and easy to follow. I also enjoyed the discussion on the limitations and future work.

**Weaknesses:**

I mainly see two weaknesses. First, while the paper is a good contribution to the robotics community, I am unsure about its fit to the ICLR venue. Overall, the learning methods applied here are not very novel. In my opinion, the main contribution of the paper lies on the hardware side. Second, I was wondering how the sim parameters were set; moreover, was a system ID of the sim parameters performed based on a small fraction of real data?

**Questions:**

See weaknesses.

---

> ### Author Response · Authors · 2025-12-01
> **Thanks to the Reviewer**
>
> We appreciate your positive assessment of our contribution to the robotics field. We certainly understand the reviewer’s perspective regarding algorithmic novelty and acknowledge the ICLR community's desire for new algorithm development. Our approach was not to propose a novel Reinforcement Learning (RL) algorithm, but rather to use existing RL methods as a powerful tool to investigate a critical aspect of embodied intelligence: the impact of physical design choices (spinal joint dynamics) on the learning process. We believe this focus on learning-based control architectures and the empirical analysis of their deep interaction with physical systems is where our work strongly aligns with ICLR’s core interest in robot learning and embodied agents.
>
>
> As noted in the paper, standard "black box" RL (SAC) converges to high-frequency, "cheetah-like" gaits in simulation that are physically infeasible on real hardware. This highlights a critical limitation in current learning-based control for non-standard morphologies. Our novelty lies in the Hybrid Control Framework. By integrating a biological prior (Hildebrand gait) to handle the rhythmic limb coordination, and relegating the RL agent to control the active spinal dynamics, we solve the sim-to-real gap by structuring of the control and policy space and constraining the RL problem in a principled way. This addresses a core question in the ICLR community: How do we effectively inject domain knowledge (priors) into learning algorithms to achieve robust real-world performance? We demonstrate that structured learning (Hybrid) succeeds where end-to-end learning fails.
>
>
> Additionally, our simulator is designed to closely match the physical salamander robot. The mechanical parameters, such as link lengths, joint placement, and mass/inertia properties, have been directly taken from the CAD model of the robot. The weights of the robot components are measured from the physical robot components; inertias are computed from CAD (with the same material densities as the real robot). The joint actuation model (torque limits, internal PID controller for servos, and damping) follows the manufacturer's specifications for the servo motors used on the robot. Regarding the ground interaction model, we used MuJoCo Menagerie [1] examples to establish a baseline and then focused on tuning the Coulomb sliding friction coefficient  within a physically plausible range; this calibration was validated by ensuring the qualitative slip/no-slip behavior observed during fundamental crawling gaits in the simulation closely matched the behavior observed on the real robot, using MuJoCo's robust default values for all other contact solver settings. We agree that implementing a formal system identification method would be a very strong approach, especially for enhancing generalizability, and we recognize its value. However, for the specific scope of this study, we found it sufficient to rely on the current configuration, which involves direct simulation using the mentioned physical model and validation through the described experimental approach. While a formal system identification would provide added rigor, we believe our current simplified method offers a practical and effective starting point when transferring our control architecture to other robotic platforms.
>
>
> Reference:
>
> [1] Zakka, K., Tassa, Y., & MuJoCo Menagerie Contributors. (2022). MuJoCo Menagerie: A collection of high-quality simulation models for MuJoCo. Retrieved from http://github.com/google-deepmind/mujoco_menagerie

---

### Note · Authors · 2026-03-09

I have read and agree with the venue's withdrawal policy on behalf of myself and my co-authors.

---

### Meta-Review · Area_Chair_PSGD · 2026-01-14

**Summary:**

This paper proposes a hybrid control framework that integrates a biologically grounded gait design with Deep RL, enabling a salamander-inspired quadruped robot to exploit active spinal joints for robust crawling motion, under environmental uncertainties such as surface irregularities.  The main concerns around ICLR fit, Novelty and lack of convincing results were not adequetely addressed during the review.

**Reviewer Concerns:**

Main concerns raised with ICLR fit (krbK, qepU,acjD), Novelty (krbK, qepU, az6p), sim2real transfer challenges / convincing results (acJD, az6P). These appear to have been inadequately addressed, e.g. the authors acknowledge that " our contribution does not lie in the novelty of the learning algorithm "

**Reviewer Scores:**

There was insufficient exchange between reviewers and authors who gave terse responses that did not fully address the concerns. It is unlikely that any of the scores would have changed.

---

### Decision · Program_Chairs · 2026-01-26

Reject